# Exercise Intervention Changes the Perceptions and Knowledge of Non-Communicable Disease Risk Factors among Women from a Low-Resourced Setting

**DOI:** 10.3390/ijerph19063474

**Published:** 2022-03-15

**Authors:** Sweetness Jabulile Makamu-Beteck, Sarah Johannah Moss, Francois Gerald Watson, Melainie Cameron

**Affiliations:** 1Physical Activity, Sport and Recreation, Faculty of Health Sciences, North-West University, Potchefstroom 2531, South Africa; sweetness.beteck@gmail.com (S.J.M.-B.); lainie.cameron@usq.edu.au (M.C.); 2Quality in Nursing and Midwifery, Faculty of Health Sciences, North-West University, Potchefstroom 2531, South Africa; francois.watson@nwu.ac.za; 3School of Health and Medical Sciences, University of Southern Queensland, Ipswich 4305, Australia

**Keywords:** Black South Africans, mixed methods, non-communicable diseases, knowledge and perceptions, physical activity, supervised exercise, Health Belief Model

## Abstract

We employed the Health Belief Model (HBM) as a theoretical lens to explore the influence of an exercise intervention on the perceptions and knowledge of modifiable risk factors for non-communicable diseases (NCDs) among women from a low-resource setting in South Africa. We used a mixed-methods design, gathering qualitative and quantitative data at baseline (n = 95) and again after 12 weeks (n = 55) and 24 weeks (n = 44) of an exercise intervention. Qualitative data consisted of focus group discussions exploring the knowledge and perceptions of modifiable risk factors for NCDs at the three time points. We collected quantitative measurements of modifiable risk factors for NCDs (waist-to-hip ratio, body mass index, blood pressure, peripheral blood glucose, and cholesterol) as well as objective physical activity (PA) data over seven consecutive days. Surveys on coronary heart disease and PA knowledge were conducted at all three time points. Qualitative findings indicated that health exposures and cultural traditions influenced the participant’s perceptions about PA and NCDs. Waist circumference significantly decreased at 12 weeks compared to baseline *M**D* = 4.16, *p* < 0.001. There was significant improvement at 12 weeks, compared to baseline, *MD* = 0.59, *p* = 0.009 for PA knowledge, and *MD* = 0.68, *p* = 0.003 for heart disease knowledge. There were reductions from baseline to 24 weeks in diastolic blood pressure (*MD* = 4.97, *p* = 0.045), waist circumference (*MD* = 2.85, *p* = 0.023) and BMI (*MD* = 0.82, *p* = 0.004). Significant heart disease knowledge improvements were found at 24 weeks compared to baseline (*MD* = 0.75, *p* < 0.001). Supervised exercise positively influenced Black African females′ health behaviours by understanding cultural perceptions of modifiable risk factors for NCDs.

## 1. Introduction

Many people do not engage in sufficient physical activity (PA) [1,2] and are at increased risk of developing non-communicable diseases (NCDs) [3,4]. NCDs are chronic diseases that are not infectious and not transmissible between people [5]. In 2012, non-communicable diseases accounted for 43% of total deaths in South Africa [6]. The most common NCDs are cardiovascular disease (CVD), cancers, respiratory diseases, type 2 diabetes, and mental health disorders [7]. Although the burden can be reduced with sufficient physical activity, in 2016, 28% of the global population were insufficiently active [2]. NCDs can be prevented and managed by engaging in PA at the recommended levels [1,4]. The World Health Organisation (WHO) recommends that the ideal level of PA for adults is 150–300 min of moderate-intensity aerobic PA, or 75–150 min of vigorous-intensity aerobic PA weekly, or an equivalent combination of moderate- and vigorous-intensity aerobic PA [8]. Studies in developed countries show that people do not know the PA recommendations to achieve health benefits [9,10,11]. Furthermore, people who have a vague understanding of the impact of PA on health tend to engage in less PA [11].

Women in sub-Saharan Africa engage in less leisure-time PA than their male counterparts. South African women also report lower PA levels than their male counterparts [12,13] and are at greater risk of developing NCDs. NCDs mainly affect women in developing countries such as South Africa [5]. African women are often overweight or obese [13,14,15,16] due to cultural acceptance of an overweight body [17,18]. To manage the risk factors of NCDs, it is essential to ascertain the PA knowledge of persons diagnosed with an NCD risk factor and how they perceive PA and how it relates to NCDs [19]. Improvements in knowledge and perceptions of NCD risk factors and PA may increase physical activity levels [20].

Studies to describe knowledge and perceptions of NCDs and PA in SA have previously been conducted [20,21,22]. A study that included deep rural, rural and urban Black Africans in SA, and which aimed to determine the relationship between NCD risk factor knowledge and risk factors, asserted that participants had good knowledge, scoring 84% for PA knowledge while most participants (67%) were leading a sedentary lifestyle [20]. Black South Africans from low-resourced environments were found to have poor perceptions about NCDs [21,22]. Intervention studies in PA aimed to improve NCD knowledge and risk factors are unavailable for a South African population.

The Health Belief Model (HBM) is one of the widely used frameworks to describe health behaviours. The HBM includes six constructs, namely, (1) perceived susceptibility, (2) perceived severity, (3) perceived benefits, (4) perceived barriers, (5) cues to action and (6) self-efficacy. The HBM was utilised as a theoretical lens [22] through which the study data were explored. Health concerns, the belief that one is vulnerable and the belief that taking action at an acceptable cost, will motivate people’s health behaviours [23,24]. People will change their behaviour when they understand the risk of the disease and know about possible solutions to mitigate the risk. Although previous studies have successfully used the HBM in analysing perceptions and knowledge of NCD risk factors among adults [21,22,25], to the best of our knowledge, no previous studies utilised the HBM in mixed-methods research to understand PA health behaviours.

We conducted this study to explore, describe and understand whether engaging women from a low-resourced community in supervised exercise could change their perceptions and knowledge of NCD risk factors and PA. By improving our understanding of the link between knowledge and perceptions and the role of supervised exercise, it will be possible to develop more sustainable PA interventions to manage NCD risk factors.

## 2. Materials and Methods

### 2.1. Design

This study is shaped by a pragmatic philosophy, which explores and describes the phenomenon. A mixed-methods study was conducted using the strength of both the qualitative and quantitative methods to yield a complete and sound analysis [26]. A convergent parallel mixed-methods design [27] was used to collect qualitative and quantitative data simultaneously at baseline, 12, and 24 weeks. Data were analysed separately, and the results were merged to explore, describe, and understand the influence of supervised exercise on the perceptions and knowledge of NCDs risk factors and PA. The HBM was applied as the lens through which the data were interpreted. This study is embedded in the B-Healthy exercise intervention study, trial no. (PACTR201609001771813), with the overarching aim to determine the influence of a controlled, supervised exercise intervention on the risk factors of NCDs, functional performance, medication usage, perceptions, and knowledge of risk factors for NCDs and quality of life in a low-resource setting.

### 2.2. Population and Sampling

The participants were volunteers recruited in 2015 via a convenience sampling technique from the Steve Tshwete Clinic catchment area in Ikageng, a dormitory town in the Dr Kenneth Kaunda District, near Potchefstroom North West Province, South Africa. The residents of the area are dependent on the public health system, report high levels of unemployment and live in informal housing with limited access to water and electricity. Based on the limited access to services, the setting is defined as low resourced. Recruitment resulted in 110 participants being assessed for eligibility for participation, of which 95 participants were included in this study. The research process flow is indicated in Figure 1 along with the measurement points. Only data from Black African women aged 28 to 78 years with at least one NCD risk factor were included in the analyses due to a minimal number of males providing informed consent. Participants that presented with a low to moderate risk, based on the seven-step physical activity readiness questionnaire (PAR-Q) that screens for risk during physical activity and reviews family history and disease severity [28], were included in this study.

### 2.3. Supervised Exercise

Participants were requested to attend supervised exercise sessions once a week for approximately sixty minutes at the Ikageng community hall. The exercise intervention was guided by the American College of Sports Medicine [29] and information from a four-week pilot study within a similar community [30]. Exercise sessions were supervised by a qualified Biokineticist (exercise physiologist) and biokinetic students-in-training and lasted sixty minutes. Participants were requested to repeat the exercise programme on two more days at their homes. Patient education was provided according to the patient education guidelines within the scope of the Biokinetics profession [31]. Resting blood pressure and exercise, heart rate, and the Borg Category-Ratio-10 scale rate of perceived exertion (RPE) [32] were recorded at each exercise session. Exercise sessions comprised a 10 min warm-up followed by 20 mins of aerobic activities at a target heart rate of 70% of age-predicted maximum heart rate reserve [29]. Fifteen minutes of resistance training with resistance bands (e.g., elastic shoulder lateral raises) and bodyweight exercises (e.g., lunges) followed the aerobic exercises (e.g., speed walking and step-ups). The session ended with 5 min of flexibility focused on large muscle groups. Progression of intensity and duration was introduced every four weeks until the aerobic activities comprised 30 min of the session. Compliance to the exercise intervention was recorded by a rollcall list and recording activity heart rates with pulse oximetry.

### 2.4. Data Collection

Qualitative and quantitative data were collected simultaneously, with qualitative data given dominance at three time points. Baseline measurements were taken and repeated after 12 weeks of supervised exercise and again after 24 weeks of supervised exercise, as shown in Figure 1. A demographic questionnaire was administered to determine the socio-economic status of the participants. The Physical Activity Readiness Questionnaire (PAR-Q) was administered to screen for participants at risk when increasing PA levels [28].

#### 2.4.1. Qualitative Data Collection

The same researchers conducted four focus group discussions (FGDs) at each measurement point. FGDs were guided using an open-ended questionnaire and based on previous research until data saturation was reached and lasted approximately an hour. Discussions included four to 10 participants per group and were conducted in English with a Tswana translator. All FGDs were audio-recorded for analyses. Baseline FGDs had a total of 34 participants. Focus group discussions consisted of 25 participants at 12 weeks and 22 participants at 24 weeks of the intervention. Qualitative data aimed to explore and describe perceptions and knowledge of NCD risk factors and PA.

#### 2.4.2. Quantitative Data Collection

Quantitative data determined the risk factors present for NCDs, knowledge of heart disease and knowledge of PA. Body composition (body mass, height, and waist and hip circumferences) measures were taken according to the guidelines of the International Society for the Advancement of Kinanthropometry (ISAK) [33]. Body mass was measured to the nearest 0.1 kg on a portable scale (Seca 876, Hamburg, Germany). Height was measured with a stadiometer (Seca 217, Hamburg, Germany) to the nearest 0.1 cm. Waist and hip circumferences were measured using non-extensible and flexible anthropometric tape (Cescorf, Porto Alegre, Brazil). Peripheral fasting blood glucose and cholesterol levels were measured using an automated device (Accutrend: Roche Diagnostics, Basel, Switzerland) according to the manufacturer’s manual [34]. Habitual PA measurements (physical activity level (PAL), activity counts, activity energy expenditure (AEE), and moderate to vigorous physical activity (MVPA) were determined objectively using a combined heart rate and accelerometry device (ActiHeart^®^, CamNtech Ltd., Cambridge, UK) [35]. The heart disease knowledge questionnaire was used to determine participants’ knowledge of a healthy diet, PA benefits, pathophysiology, and risk factors relevant to heart disease and heart attack symptoms [36]. The PA knowledge questionnaire was previously used in a similar community in the North West Province to determine knowledge of PA in health improvement [37]. The Cronbach’s alpha coefficient for the PA knowledge questionnaire was 0.66 for questionnaire reliability.

### 2.5. Data Analysis

#### 2.5.1. Qualitative Data Analysis

Qualitative data analyses were performed at baseline, 12 weeks, and 24 weeks of exercise intervention. Audio recordings were transcribed verbatim. Transcripts were analysed through inductive content analysis [38], utilising a non-linear approach of “noticing, collecting and thinking about it” (NCT approach) [39]. This process was carried out with the assistance of ATLAS.ti 8 for Windows [40], computer-assisted qualitative data analysis software (CAQDAS). Through data saturation [41], an exhaustive overview was drawn based on participants’ perceptions of NCDs and PA through the theoretical lens of the HBM.

#### 2.5.2. Quantitative Statistical Analysis

The Statistical Package for Social Sciences (SPSS) ver. 26.0 software (IBM SPSS Statistics, Chicago, IL, USA) was used for statistical analyses. The Shapiro–Wilk test was used to determine normality for the standardised residuals of outcome measures. Descriptive analyses were performed, reporting means and standard deviations, medians, inter-quartile range, and frequencies. For normally distributed data, repeated-measures ANOVA was performed to determine differences in the risk factors for NCDs, knowledge of PA and knowledge of NCDs at baseline, three and six months post the supervised exercise intervention. Greenhouse–Geisser or Huynh–Feldt correction was used when Mauchly’s test of sphericity was violated. The effect size was determined by running partial eta squared $(\eta_{p}^{2}$) and is classified as small if $\eta_{p}^{2}=0.01$, medium if $\eta_{p}^{2}\;$= 0.06 and large if $\eta_{p}^{2}$ = 0.14. Post hoc tests with Bonferroni correction were conducted to determine the differences for parametric data. Friedman’s test was conducted to compare for non-normally distributed outcome measures. Dunn–Bonferroni post hoc tests were conducted to determine where the differences were. Kendall’s W (coefficient of concordance) was carried out to determine the level of agreement for non-normally distributed data. The effect size is considered small if 0.1 ≤ W < 0.3, moderate if 0.3 ≤ W < 0.5, and large if W ≥ 0.5. The level of significance was set at *p* ≤ 0.05.

#### 2.5.3. Mixed-Methods Analysis

A side-by-side comparison was employed to identify the convergent and divergent findings within the qualitative and quantitative data [42].

### 2.6. Rigour

To ensure consistency, the researcher, proficient in Tswana, engaged all focus groups in the participants’ environment until saturation [43] was reached. Focus group discussions were held in English with a Tswana translator to ensure full participation. The researcher, experienced in conducting FGDs, also analysed the data [44] to promote credibility. A more experienced researcher, the supervising researcher, independently analysed the transcripts. Through peer debriefing [45], the two researchers discussed their findings for consensus [46].

### 2.7. Ethical Considerations

This study was approved by the Health Research Ethics Committee for Humans at the North-West University (NWU 000049-15-A1-03) and the North West Department of Health. Information sessions were held with potential participants, where they received information sheets. Written consent was obtained from all participants. Partial confidentiality was ensured for FGDs, and all data were collected according to the approved protocol and the guidelines of the Helsinki Declaration [47].

## 3. Results

### 3.1. Characteristics of the Study Participants

A total of 95 Black South African women aged 28 to 78 years, with a mean age of 56 ± 12.6 years, formed part of the B-Healthy exercise intervention study. A total of 42% of the participants dropped out from baseline to 12 weeks. A total of 20% of participants dropped out from 12 weeks to 24 weeks. The women come from impoverished families, with household income less than R100,000 per annum (82.9%). These women reported low education levels, with only 3.2% reporting college qualifications. Only 38.9% of women were married, and the rest were single or widowed. The characteristics of the study participants are presented in Table 1.

The HBM [48] and its constructs for changing perceptions and knowledge of NCDs and PA were employed to present and analyse the results. Qualitative explorations and quantitative determinants were merged through a side-by-side comparison to explore and describe knowledge and perceptions and establish the constructs of the HBM as a theoretical basis.

### 3.2. Qualitative Findings

Results that encapsulate the broad pre-determined themes, “perceptions of NCDs and perceptions of PA” from the qualitative investigation, are presented in Table 2.

The qualitative findings revealed that, at baseline, participants perceived that non-modifiable risk factors increase the risk of developing NCDs. Participants perceived that they could not prevent NCDs since they are considered genetic conditions. Most participants expressed a lack of knowledge of NCD risk factors. Participants also conveyed a lack of knowledge about the types of PA they could engage in. The participants were aware of the role of PA in health. Fear of injury posed a barrier to PA engagement, particularly the lack of knowledge of safe activities.

At 12 weeks, participants reflected on the past to improve their ability to understand why they were diagnosed with an NCD. They perceived that changes in diet (from traditional African to modern Westernised) increased NCDs. Participants described disease complications to indicate that NCDs are severe conditions, and they were aware of NCDs through personal diagnoses. The results suggest that medical professionals positively raise awareness of adopting PA as part of a management plan for NCDs. Walking was the typical PA that most participants mentioned, preferably in the morning hours. Though unable to fully describe the benefits of PA to health, participants enjoyed participating in PA within a group setting.

At 24 weeks, the participants reported that the modern diet increases the risk of developing NCDs. Their perceptions were that NCDs are severe conditions that could result in loss of life. The results suggest that participants have limited knowledge about NCDs. They are aware of disease-management skills, including dietary advice they receive at the primary health clinics. The preferred time to engage in PA is in the morning hours. The results indicate that experiential knowledge through group exercises increased the participants′ knowledge of the type of PA they could engage in. Participants did not perceive barriers to exercise, and they believed that PA forms part of disease management.

### 3.3. Quantitative Findings

#### 3.3.1. Prevalence of NCD Risk Factors

Risk factors for NCDs were rife among participants, including unfavourable body composition and hypertensive blood pressure, as shown in Table 3. Many participants presented with increased risk factors for NCDs, including BMI levels above 30 kg/m^2^, waist-to-hip ratios greater than 0.86, SBP above 130 mmHg, DBP above 80 mmHg, blood glucose ≥5.6 mmol/L and total cholesterol ≥5.2 mmol/L, as shown in Figure 2.

The prevalence of large waist circumferences (82%) was highest at baseline. Diastolic blood pressure (73%) was the second most prevalent risk factor at baseline. Over half (56%) of the participants were obese at baseline and 12 weeks. The prevalence of large waist circumference reduced by 13%, to 69% at 12 weeks, and stabilized at 24 weeks (73%). Diastolic blood pressure reduced at 12 weeks with 62% of the participants reporting elevated diastolic blood pressure. The elevated systolic blood pressure prevalence reduced by 8% at 12 weeks, accounting for 55% and stabilised at 24 weeks, accounting 59%. The prevalence of obesity reduced at 24 weeks from 56% to 50%. The prevalence of diastolic blood pressure was also reduced to 46%. The prevalence of glucose (27%) and total cholesterol (17%), though low at baseline, increased at 12 weeks, accounting for 36% for blood glucose and 21% for total cholesterol. The prevalence of blood glucose further increased at 24 weeks, accounting for 42%, while total cholesterol decreased by 2% at 24 weeks, accounting for 19%.

#### 3.3.2. NCD Risk Factors over Time

The changes in NCD risk factors from baseline to 12 weeks and 24 weeks are presented in Table 3.

The repeated-measures ANOVA with a Huynh–Feldt correction showed that weight significantly differed between the measurement points, F (1.71, 68. *p* = 0.030,$\;\eta_{p}^{2}$ = 0.09 (see Table 3). Post hoc tests with Bonferroni correction showed that weight changes were not significant between baseline and 12 weeks (*p* = 0.077) and between 12 weeks and 24 weeks (*p* = 0.064).

In the case of BMI, repeated-measures ANOVA results using a Huynh–Feldt correction indicate significant differences between the time points, F (1.61,64.52) = 8.60, *p* < 0.001, $\eta_{p}^{2}$ = 0.18. Post hoc tests with Bonferroni correction showed significantly lower BMI at 24 weeks than baseline (*p* < 0.001).

There was a significant difference in the measurement points for waist circumference, F (2, 80) = 11.72, *p* < 0.001, $\eta_{p}^{2}$ = 0.23. Post hoc pairwise analyses using Bonferroni correction were significant between baseline and 12 weeks (*p* < 0.001) and baseline to 24 weeks (*p* = 0.023) but not significant between 12 and 24 weeks (*p* = 0.247).

Repeated-measures ANOVA using Greenhouse–Geisser correction show that waist-to-hip ratio significantly differed between the time points, F (1.47, 58.91) = 11.66, *p* < 0.001, $\eta_{p}^{2}$ = 0.23. Post hoc tests with Bonferroni correction showed that participants had a significantly lower waist-to-hip ratio at 12 weeks compared to baseline (*p* < 0.001) and at 12 weeks compared to 24 weeks (*p* = 0.010).

In the case of diastolic blood pressure (DBP), there was a significant difference over time, F (2, 80) = 4.10, *p* = 0.020, $\eta_{p}^{2}$ = 0.093, as shown in Table 3. Post hoc tests with Bonferroni correction show that DBP was significantly lower at 24 weeks than baseline (*p* = 0.045) but not between baseline and 24 weeks (*p* = 0.208).

There was no significant effect on objective measurements of total free daily living PA across the time points. Participants spent more time in MVPA at 12 weeks (57.24 min/day [0–757.0]) compared to baseline (43.97 min/day [0–437.0]) and 24 weeks (44.84 min/day [0–175.0]) (see Table 3).

Friedman’s test shows that heart disease knowledge significantly affected the time points ꭓ^2^(2) = 14.56, *p* < 0.001, W = 0.18). Pairwise comparison with Dunn–Bonferroni correction shows significant differences between baseline and 12 weeks *p* = 0.003 and baseline and 24 weeks *p* < 0.001. There was no significant difference between 12 and 24 weeks, *p* = 1.00. The findings indicate an improvement in heart disease knowledge at 12 weeks and 24 weeks after baseline testing, as shown in Table 3.

Friedman’s test shows that PA knowledge scores differed significantly across the time points (ꭓ^2^(2) = 9.62, *p* = 0.010, W = 0.12). Post hoc analyses with Dunn–Bonferroni correction show a significant difference between baseline knowledge of PA compared to 12 weeks (*p* = 0.009). There was no significant difference between baseline and 24 weeks or between 12 and 24 weeks after baseline testing.

#### 3.3.3. Knowledge of Risk Factors for NCDs and PA

At baseline, participants lacked knowledge about NCD risk factors, scoring an average of 39% for the coronary heart disease knowledge survey, as shown in Table 4. Age is a non-modifiable risk factor for NCDs; most participants could not associate heart disease risk with increasing age. At 12 weeks, there were improvements in the coronary heart disease knowledge survey. More participants knew that heart disease is better defined as a chronic long-term illness than a short-term illness; participants scored an average of 71% for the questionnaire. At 24 weeks, more than half of the participants knew that high blood pressure is not defined as 110/80 mmHg (systolic/diastolic) or higher—they scored on average 56% for the questionnaire.

The PA knowledge survey results indicated that most of the participants knew the effect of PA on health, scoring high on PA knowledge, as shown in Table 5. At 12 weeks and 24 weeks, fewer people knew that PA is suitable for all individuals compared to 12 weeks and baseline.

### 3.4. Mixed-Methods Findings

Qualitative and quantitative results are integrated and presented in Table 6. At baseline, qualitative assertions and quantitative findings indicate that participants lack knowledge about NCDs and risk factors. Participants knew about PA but perceived an inability to engage in PA. Participants lack knowledge about different types of PA that they can safely engage in. Exposures to PA through supervised exercise improved knowledge of PA at 12 weeks. Decreases in NCD risk factors and motivation to exercise from other participants improved participants′ self-efficacy for PA engagement. At 24 weeks of supervised exercise, participants perceived the ability to engage in PA and expressed fewer barriers for PA engagement. Experiential knowledge improved their knowledge and perceptions of PA and NCDs.

## 4. Discussion

This mixed-methods study used the HBM as the lens to explore and describe perceptions and knowledge of NCD risk factors and PA among women from a low-resourced community. Black South Africans tend to lack knowledge and have negative perceptions about NCDs and PA [21,22,25]; This study aimed to understand the influence of a 24 week supervised exercise programme on perceptions and knowledge of NCDs and PA. Before engaging in a 24 week supervised exercise intervention, participants perceived low self-efficacy relating to NCD prevention. This may be related to low education levels and low health literacy [49]. Many participants were aware of the terminology used for NCD conditions through personal diagnoses but did not understand NCDs. This finding agrees with assertions made by Surka et al. [50], who investigated knowledge and perceptions of cardiovascular disease risk factors among adults from a low-income, peri-urban community in the Western Cape Province of South Africa and found that respondents were knowledgeable about cardiovascular disease terminology.

In this study, participants initially perceived an increased risk of NCDs was caused by non-modifiable risk factors, including family history and age. According to Sallis [51], family history contributes approximately 20% to overall health. Contrary to the perceptions of the participants in this study, modifiable risk factors including alcohol abuse, smoking, unhealthy diet, and physical inactivity are the major causes of NCDs. In South Africa, being female (odds ratio 18.23; CI 16.75–19.85) and aged 59 to 77 years (odds ratio 1.37; CI1.24–1.50) or 78 to 98 years (odds ratio 1.25; CI 1.16–1.35) is associated with stroke prevalence. Diabetes (odds ratio 14.53; CI 13.36–15.79) and heart disease (odds ratio 8.86; CI 8.23–9.55) are associated with stroke. The 12-week supervised exercise was associated with increased awareness of non-modifiable risk factors for NCDs. Though not fully able to grasp the role of adopting healthy lifestyle behaviours to prevent NCDs, participants perceived that their traditional diet protected them against NCDs. More than half of the participants had unhealthy body composition [52] and high blood pressure [13]. Even though most were already on prescribed anti-hypertensive medication, high blood pressure was prevalent. Consistent with a previous study among South Africans, this trend of uncontrollable hypertension may be due to sedentary lifestyles or low-to-moderate PA [53]. Body composition and blood pressure improvements were significant at 12 weeks of supervised exercise in this study. A similar study consisting of a four-week exercise intervention conducted in a low-resourced community in Potchefstroom found (2% to 3%) improvements in NCD risk factors in just four weeks, though coupled with a participation drop-out of 29% from baseline measurements owing to PA barriers among 26 men and 50 women aged 35 to 65 years [31].

Though participants were aware, through media, of various types of PA, including sporting activities, they did not see themselves engaging in such activities because of fear of injury. Participants were generally knowledgeable that PA is good for health. However, they lacked knowledge about appropriate PA activities they could engage in. This finding is consistent with an observational study in a similar setting where participants reported enjoying walking and house chores as their preferred form of PA [54]. Participants expressed enjoyment in engaging in PA and preferred readily accessible forms of PA such as walking, house chores, gardening, and group exercise. The preferred time of the day for PA engagement is early-morning hours. After 24 weeks of exercise, participants particularly expressed that they enjoyed group exercise to increase PA. Silva et al. [55] asserted that people do not look merely for the health benefits of PA (that is, reducing NCDs and death), but they look for happiness and social value. Barriers to PA engagement include South Africa’s high crime rate (for instance, the dangers of outdoor exercise) and lack of infrastructure in low-resourced areas. Smit et al. report that efforts to increase PA, such as walking, among Black South Africans are hampered by the lack of public space for outdoor activities and the high crime rate [56].

Consistent with the findings by Sehole and Van der Heever [57], participants in this study expressed a lack of knowledge of NCDs and NCD risk factors. Engaging these women in the 24 week exercise programme improved their perceptions about NCD risk factors and PA. There were significant improvements in BMI, waist circumference, DBP, and coronary heart disease knowledge following 12 weeks of supervised exercise and raised awareness of PA’s role in NCD risk factor management. This finding is supported by Dorman et al. [58], who asserted that individuals at risk of NCDs due to their health profile often fail to perceive a threat. Engaging in PA as part of a group was most often preferred. The findings of this study have to be interpreted against the limitations observed, including the high dropout rate. The reason may be transport-related barriers to attending the interventions or finding a permanent job. The intervention was administered during working hours, which might have encouraged dropout due to availability during the day. The 12-week exercise was successful in increasing daily PA levels among African women. Persons who do not engage in PA often need experiential knowledge of PA, not only education. This is supported by Hay-Smith et al. [59].

## 5. Conclusions

During the 24 weeks of supervised exercise interventions, a positive effect was found on the NCD risk factors and PA knowledge. Although physical improvements were observed during the 12 weeks of the supervised exercise intervention, perceptions concerning risk factors of NCD and PA changed only after 24 weeks of the exercise intervention. A supervised exercise intervention seems to positively influence women’s health behaviours from a low-resourced community through a deeper understanding of the influence of cultural perceptions on modifiable risk factors for NCDs. Interventions should preferably be longer than 12 weeks to change perceptions and planned outside working hours in a safe environment to reduce program attrition.

## Figures and Tables

**Figure 1 ijerph-19-03474-f001:**
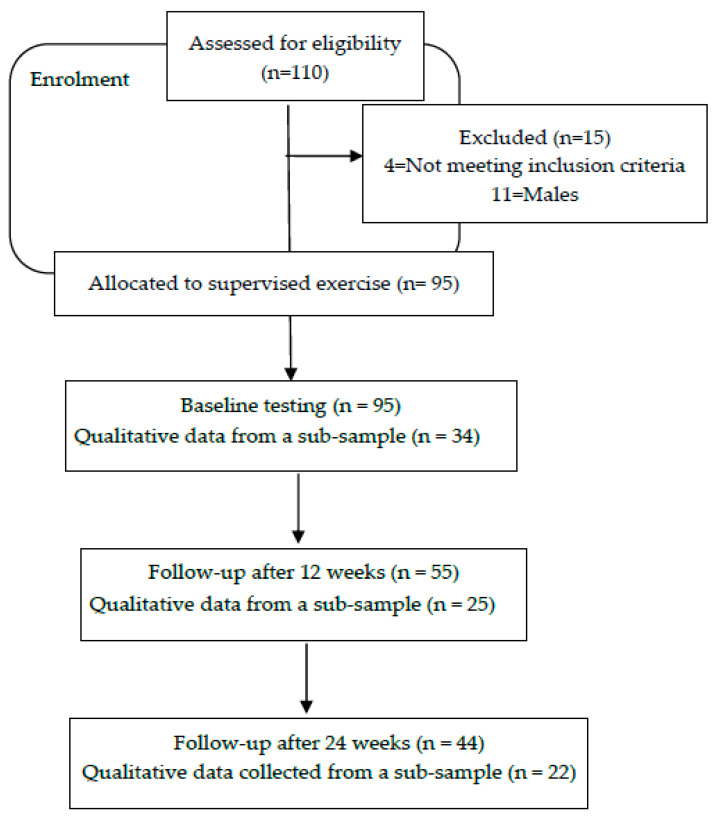
Flow diagram of study participants in the exercise intervention indicating the sample size of the qualitative and quantitative arms of this study.

**Figure 2 ijerph-19-03474-f002:**
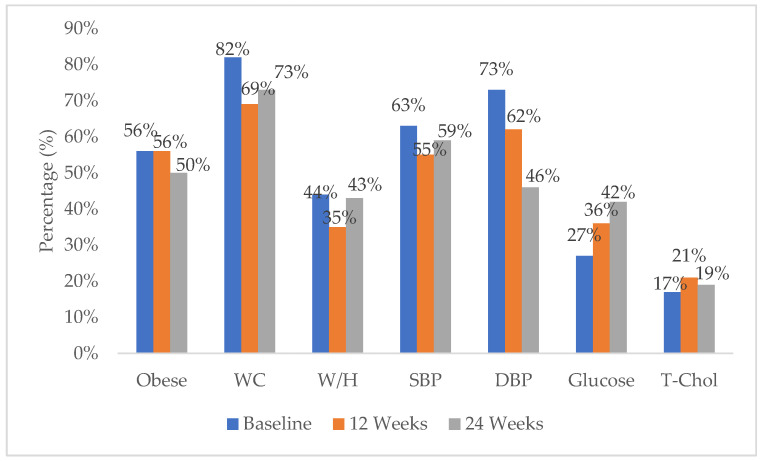
The percentage of participants presenting with non-communicable disease risk factors at baseline, 12 weeks, and 24 weeks of exercise intervention. Note. WC = waist circumference, W/H = waist-to-hip ratio, SBP = systolic blood pressure, DBP = diastolic blood pressure, and T-chol = total blood cholesterol.

**Table 1 ijerph-19-03474-t001:** Characteristics of study participants.

Variable	Number (%)
Socio-economic variables	
Highest level of education:	
No schooling	35%
High school	61%
Diploma	3%
Employment status	
Employed	17%
Unemployed	59%
Unable to work or retired	24%
Marital status	
Married	39%
Single	38%
Widowed	21%
Persons in household	
1–3	33%
4–6	54%
>6	14%
Household income	
Less than R100,000	83%
R100,000–R250,000	9%
R250,000–R400,000	8%

**Table 2 ijerph-19-03474-t002:** Qualitative findings.

Theme	Category	Sub-Category	Quotation
**Baseline**			
**Individual perceptions**	Increased risk	Genetics	Participant D3_01: “You get NCDs among relatives who stay together in the same house; you can’t prevent NCDs as they are hereditary”
**Modifying factors**	Knowledge of NCDs	Lack of knowledge about NCDs	Participant D3_01: “I don’t understand diabetes. Isn’t diabetes caused by the lack of vitamins in the body?”
	Knowledge of PA	Lack of knowledge about PA	Participant D1_01: “I don’t know what other push-ups there are”
**Likelihood of action**	Benefits of PA	PA lowers fat in the heart	Participant D2_01: “PA makes your heart not to be full of fat”
	Barriers to PA	Fear of injury	Participant D1_01: “On the TV grannies that play soccer, I always saw them; they kick the ball and score. I’m ‘oh my God, I wish I was also there’. So now [I am] here because my knees are so painful, looks like they will break”
**12 weeks**			
**Individual perceptions**	Increased risk	Modern diet	Participant D6_01: “Milo, fruit and wild berries were like medicine. When we stopped eating those things, we started having high blood pressure and diabetes”
	Seriousness of NCDs	Disease complications	Participant D6_01: “When you have hurt your leg and have a sore, it doesn’t heal; you will go to amputate the leg. That is what I understand about diabetes”
**Modifying factors**	Knowledge of NCDs	Personal diagnoses	Participant D8_01: “I was told I have heart failure at the hospital”
	Knowledge of PA	Brisk walking	Participant D8_01: I was pacing every morning as the doctor had instructed. When I went for the check-up, the doctor said my heart is fine”
	Cues to action	Encouragement from medical professionals	Participant D6_01: The doctor spoke that we should keep on gyming because our diabetes is fine”
**Likelihood of action**	Benefits of PA	Enjoyment	Participant D5_03: “The exercise group is fun; it helps with something”
**24 weeks**			
**Individual perceptions**	Increased risk	Modern diet	Participant D10_01: “The food we eat these days causes us to be prone to NCDs”
	Seriousness of NCDs	NCDs are life threatening	Participant D12_01: “NCDs are dangerous diseases. When you have any NCD, you know your life is in danger”
**Modifying factors**	Knowledge of NCDs	Lack of knowledge about NCD risk factors	Participant D10_01: “I don’t know what diabetes is, but I was told I have a disease called diabetes, they said what kind of food we should eat”
	Knowledge about PA	Exercise group encourages capacity building	Participant D12_01: “My husband leaves at 5 am to go to work. When he leaves, I get up from the blankets, go to the sitting room and do the exercises we do in the exercise group”
	Cues to action	Awareness through media	Participant D9_01: “I heard on the radio they said eating and sitting and a child that does not play, watching TV the whole day, *gets kidney disease and diabetes”*
**Likelihood of action**	Benefits of PA	Disease control	Participant D11_01: “When you are active, high blood pressure becomes level”

**Table 3 ijerph-19-03474-t003:** Risk factors for non-communicable diseases at baseline, 12 weeks, and 24 weeks of supervised exercise in a low-resourced community.

NCD Risk Factor	N	Baseline M ± SD	12 Weeks M ± SD	24 Weeks M ± SD	*p*	$\mathrm{Effect}\;\mathrm{Size}{\;}^{1}\;(\eta_{p}^{2})\;\mathrm{or}\;W\;$
Weight (kg)	41	76.2 ± 19.2	75.4 ± 19.6	74.9 ± 19.8	0.03	0.09
BMI (kg/m^2^)	41	32 ± 7.8	31 ± 7.9	31 ± 7.8	<0.001	0.18
Waist (cm)	41	92.7 ± 13.33	88.6 ± 12.99	89.9 ± 13.04	<0.001	0.23
W/H ratio	41	0.87 ± 0.92	0.83 ± 0.76	0.84 ± 0.67	<0.001	0.23
SBP (mmHg)	41	140 ± 16	133 ± 13	135 ± 13	0.08	0.06
DBP (mmHg)	41	85 ± 13	82 ± 11	80 ± 11	0.02	0.09
Glucose (mmol/L)	37	5.0 [1.2–12.3]	6.3 [1.6–21.5]	5.7 [2.9–15.0]	0.09	0.06
T-chol (mmol/L)	37	4.1 ± 1.3	4.2 ± 9.6	3.9 ± 1.3	0.21	0.04
PAL	27	1.40 ± 0.19	1.55 ± 0.53	1.45 ± 0.24	0.79	0.01
Activity (counts/min)	27	19.73 ± 13.26	20.65 ± 8.54	17.69 ± 11.40	0.17	0.07
AEE (kcal/week)	27	448.44 [31.00–2219.00]	503.47 [23.00–3439.00]	421.05 [1.00–1075]	0.86	0.04
MVPA (min/day)	27	43.97 [0–437.0]	57.24 [0–757.0]	44.84 [0–175.0]	0.35	0.04
ActiHeart® (days worn)	27	6 ± 1	6 ± 1	6 ± 2	0.37	1.00
Heart disease knowledge (n)	40	12 [2,3,4,5,6,7,8,9,10,11,12,13,14,15,16,17,18,19,20]	14 [2,3,4,5,6,7,8,9,10,11,12,13,14,15,16,17,18,19]	14 [7,8,9,10,11,12,13,14,15,16,17,18,19,20]	<0.001	0.18
Physical activity knowledge (n)	40	9 [5,6,7,8,9,10]	9 [6,7,8,9,10]	9 [6,7,8,9,10]	0.01	0.12

^1^ Partial eta squared were used for repeated-measures ANOVA, effect size W was used for non-parametric data. BMI—body mass index; W/H—waist-to-hip ratio; SBP—systolic blood pressure; DBP—diastolic blood pressure; T-chol—total serum cholesterol; PAL—physical activity level; AEE—activity energy expenditure; MVPA—moderate-to-vigorous physical activity.

**Table 4 ijerph-19-03474-t004:** Number and percentage of participants scoring correct answers for the heart disease survey at the three time points.

Survey Questions	Baseline (n = 95) n (%)	12 Weeks (n = 55) n (%)	24 Weeks (n = 43) n (%)
Polyunsaturated fats are healthier for the heart than saturated fats.	21 (22)	35 (64)	18 (42)
Women are less likely to get heart disease after menopause than before.	10 (11)	18 (33)	14 (33)
Having had chickenpox increases the risk of getting heart disease.	24 (25)	13 (24)	21 (49)
Eating a lot of red meat increases heart disease risk.	50 (53)	35 (64)	28 (65)
Most people can tell whether or not they have high blood pressure.	31 (33)	9 (16)	16 (37)
Trans-fats are healthier for the heart than most other kinds of fats.	12 (13)	4 (7)	4 (9)
The most important cause of heart attacks is stress.	4 (4)	2 (4)	1 (2)
Walking and gardening are considered types of exercise that can lower heart disease risk.	87 (92)	55 (100)	40 (93)
Most of the cholesterol in an egg is in the white part of the egg.	23 (24)	28 (51)	11 (26)
Smokers are more likely to die of lung cancer than heart disease.	5 (5)	4 (7)	1 (2)
Taking an aspirin each day decreases the risk of getting heart disease.	44 (46)	16 (30)	29 (67)
Dietary fibre lowers blood cholesterol.	66 (70)	52 (95)	27 (63)
Heart disease is the leading cause of death in the United States.	64 (67)	32 (58)	27 (63)
The healthiest exercise for the heart involves rapid breathing for a sustained period.	56 (60)	26 (47)	17 (40)
Turning pale or grey is a symptom of having a heart attack.	32 (34)	21 (38)	24 (56)
A healthy person′s pulse should return to normal within 15 min after exercise.	59 (62)	35 (64)	27 (63)
Sudden trouble seeing in one eye is a common symptom of heart attack.	39 (41)	64 (35)	25 (58)
Cardiopulmonary resuscitation (CPR) helps to clear clogged blood vessels.	7 (7)	19 (16)	9 (21)
HDL refers to “good” cholesterol, and LDL refers to “bad” cholesterol.	43 (45)	11 (20)	11 (26)
Arterial defibrillation is a procedure where hardened arteries are opened to increase blood flow.	10 (11)	5 (9)	7 (16)
Feeling weak, lightheaded, or faint is a common symptom of having a heart attack.	47 (50)	29 (53)	25 (58)
Taller people are more at risk of getting heart disease.	47 (50)	35 (64)	28 (65)
“High” blood pressure is defined as 110/80 (systolic/diastolic) or higher.	32 (34)	19 (35)	24 (56)
Most women are more likely to die from breast cancer than heart disease.	4 (4)	5 (9)	9 (21)
Margarine with liquid safflower oil is healthier than margarine with hydrogenated soy oil.	38 (40)	38 (69)	24 (56)
People who have diabetes are at higher risk of getting heart disease.	64 (67)	47 (86)	26 (61)
Men and women experience many of the same symptoms of a heart attack.	44 (46)	31 (56)	26 (61)
Eating a high fibre diet increases the risk of getting heart disease.	34 (36)	41 (75)	26 (61)
Heart disease is better defined as a short-term illness than a chronic, long-term illness.	36 (38)	39 (71)	24 (56)
Many vegetables are high in cholesterol.	75 (79)	50 (91)	30 (70)
**Tot Average percentage of correct scores**	**39%**	**45%**	**47%**

**Table 5 ijerph-19-03474-t005:** Number and percentage of participants who scored correct answers for the physical activity survey at baseline, 12 weeks, and 24 weeks post supervised exercise intervention.

Item	Baseline (n = 95) n (%)	12 Weeks (n = 55) n (%)	24 Weeks (n = 43) n (%)
Physical activity is only suitable for some individuals, e.g., elite sports people/young people/Caucasians	59 (62)	46 (84)	24 (56)
Exercise reduces high blood glucose (sugar) levels/diabetic complications	58 (61)	52 (95)	42 (98)
Physical activity of moderate intensity at least five times a week has positive effects on health	90 (95)	55 (100)	40 (93)
Exercise decreases physical dependence	66 (70)	49 (89)	37 (86)
Thirty minutes of physical activity everyday supports weight loss	84 (88)	51 (93)	39 (91)
Physical activity is good for your blood pressure no matter your age, weight, race, or gender	90 (95)	52 (93)	39 (91)
Physical activity causes/worsens pain	75 (79)	48 (87)	39 (91)
Exercise contributes to cholesterol control	90 (95)	53 (96)	43 (100)
Physical activity contributes to a better state of mind	92 (97)	55 (100)	43 (100)
Physical activity improves health and general wellbeing	93 (98)	55 (100)	43 (100)
**The average percentage of correct scores**	**87%**	**94%**	**91%**

**Table 6 ijerph-19-03474-t006:** Integration of qualitative and quantitative results.

Health Belief Model Constructs	Integrated Qualitative and Quantitative Results
**Baseline**	
**Individual perceptions**	Qualitative and quantitative data analyses show that participants’ perceptions of NCDs correspond with measured knowledge of NCD risk. Participants perceive **increased risk to NCDs** to be related only to non-modifiable risk factors. Scores on heart disease knowledge were lowest on medical and risk factors.
**Modifying factors**	NCD risk factors are prevalent among participants; however, participants demonstrate a lack of **knowledge about NCDs** risk factors. Although good **knowledge of PA**, participants perceive a lack of knowledge about types of PA.
**Likelihood of action**	Perceived **benefits of PA** include a reduction in fat in the heart. Participants mainly engaged in low-intensity PA, which may be related to the perceived fear of injury, a **barrier to PA**.
**12 Weeks**	
**Individual perceptions**	Participants perceived **increased risk of NCDs** because of dietary changes from previous generations. Participants were generally knowledgeable about dietary risk factors for heart disease. Referring to disease complications, participants perceived that **NCDs are serious** conditions.
**Modifying factors**	A significant decrease in some NCD risk factors was coupled with improvements in heart disease knowledge. Participants reported **knowledge of NCDs** through personal diagnoses by medical professionals. Participants had more **knowledge about PA**. Medical professionals raised awareness for PA engagement. Participants knew about readily accessible forms of PA.
**Likelihood of action**	The enjoyment of PA was perceived to be a **benefit of PA**. In particular, PA engagement within a group. Objective PA measurements show that time spent in PA improved. Participants did not perceive any **barriers to PA** engagement. Surprisingly, they were not knowledgeable about the benefits of PA in NCD management.
**24 Weeks**	
**Individual perceptions**	Participants perceived that the modern diet **increases the risk of NCDs**, and NCDs are **serious conditions** because they are life threatening.
**Modifying factors**	Participants show improvements in heart disease knowledge. On the other hand, they perceive a lack of **knowledge about NCDs**, though they are aware of the disease conditions through personal diagnoses. There were significant decreases in most NCD risk factors from baseline, including BMI and diastolic BP. Media campaigns raised awareness of the connection between PA and NCDs. **PA knowledge** improvements were seen. Participants had experiential knowledge of the types of exercises they could do.
**Likelihood of action**	Perceived **benefits of PA** included disease management. Participants did not perceive **barriers to PA**.

## Data Availability

All data are available upon request from the corresponding author.

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
