# Peer review of "Exercise Intervention Changes the Perceptions and Knowledge of Non-Communicable Disease Risk Factors among Women from a Low-Resourced Setting"

_ijerph, 2022, doi:10.3390/ijerph19063474_

Round 1

Reviewer 1 Report

My comments below are numbered according to the order of feedback comments from the authors.

  1. The manuscript title has been improved by eliminating the “supervised exercise” term. However, the manuscript title you listed in the feedback comments is not the same as the title shown in the revised manuscript. The title in the revised manuscript is not grammatically correct. The correct title should be “Exercise intervention changes the perceptions and knowledge of noncommunicable disease risk factors among women from a low-resource setting.

           There is an additional problem with the title. What is a “low resource setting” because you did not define or measure this variable in the paper? I see nothing in section 2.2 (Population and sampling) about a low resource setting and how you measured this with your participants. If you measured this variable, please define it and provide the specific criteria and data you used to make this assertion. Otherwise, please delete “low resource setting” from the title.

  1. The revision is acceptable.

  1. Ok, this is clearer. The revision is acceptable.

  1. Ok, this is clearer. The revision is acceptable.

  1. Ok, this is clearer. The revision is acceptable

  1. Ok, this is clearer. The revision is acceptable.

  1. Ok, this is clearer. The revision is acceptable.

  1. Ok, this is clearer. The revision is acceptable.

  1. Can you explain which RPE scale you used (with citation) and what defined moderate and vigorous?

  1. Ok, this is clearer. The revision is acceptable

  1. Ok, this is clearer. The revision is acceptable.

  1. Ok, this is clearer. The revision is acceptable. I also see that on lines 200-201, you discussed saturation. My apologies if I missed this before.

  1. Ok, this is clearer. The revision is acceptable.

  1. Ok, so you did perform an intention-to-treat analysis (just not the LOCF method). This is fine. After further consideration, I agree with your rationale for using an RM-ANOVA.

  1. Ok, this is clearer. Thank you for clarifying. Also, I’m glad to see that you mentioned using the ATLAS.ti 8 software. The revision is acceptable.

  1. The revision is acceptable.

  1. Perhaps I missed this before. Again my apologies. I see the p values in the text, as you described.

  1. Ok, this is clearer. The revision is acceptable.

Additional minor corrections needed:

-Figure 1 title needs to be capitalized consistently according to APA guidelines

-line 145, please correct “ana-lyses”; it should be “analyses.”

-Table 1, Table 2, Table 3, Table 4, Table 5, and Table 6 titles need to be capitalized (and formatted) consistently according to APA guidelines

Reviewer 2 Report

The manuscript entitled "Exercise intervention changes women from a low-resource setting’s perceptions and knowledge of non-communicable disease risk factors" uses the Health Belief Model to understand the influence of exercise intervention among women from a low-resource setting in South Africa. The Authors introduce their research problem in the introduction, which is well structured, but the "Material and Methods" section needs minor changes. Please describe PAR-Q. How many items does PAR-Q have? What are the answer categories, etc? Furthermore, I recommend using examples from your questionnaire, especially on qualitative measures. The results are well presented. I have recommendations for the discussion to the authors. First, try to use more references that could explain your result and put your result into global context in this part. I also recommend including a self-administer questionnaire as a limitation since most of the questionnaires are not 100% objective.

Author Response

This manuscript is a resubmission of an earlier submission. The following is a list of the peer review reports and author responses from that submission.

Round 1

Reviewer 1 Report

Title: The Influence of Supervised Exercise on Perceptions and Knowledge of Non-Communicable Disease Risk Factors and Physical Activity among Women from a Low Resourced Setting: A Mixed-Methods Study

Developing a theory-based behavior change intervention for physical activity to improve health is potentially a useful research endeavor. This paper aimed to investigate the implementation of a behavior change intervention based on the Adapted Physical Activity Health Belief Model (PAHBM) to improve the efficacy of an exercise intervention.

With that said, however, I have several concerns, which I have tried to outline below. Most of the issues are areas where further details and explanations are needed.

Title, Intro & Purpose:

  • The title of the study contains the words “supervised exercise” yet, a majority of the exercise sessions were unsupervised (lines 103-104 state “two or more days at their homes”). Only one session was supervised (lines 100-101). The title is therefore misleading, and it should be reworded.
  • The abstract mentioned “Adapted Physical Activity Health Belief Model (PAHBM),” yet this theory is not discussed in the introduction. Lines 63-70 briefly mention the “Health Belief Model” but not the “Adapted Physical Activity Health Belief Model.” Furthermore, minimum information is provided on the Health Belief Model and its components (e.g., perceived severity, perceived susceptibility, perceived benefits, perceived barriers, and cues to action). This section needs to be expanded considerably and include specific references to the exercise context and the theory's efficacy in changing behavior.
  • I am not sure I understand the purpose of the study. Lines 71-75 state that simply by exercising, the participants will perceive and understand their risk factors? Isn’t it the case that you are implementing a behavior change intervention that educates and informs participants on their risks, severity, barriers, benefits, and cues to action (within the framework of an exercise program) in order to increase adherence and compliance to the program?
  • The introduction mentioned risks for developing non-communicable diseases, but you did not mention or discuss the different types of non-communicable diseases and their relevance to the population you are studying or how exercise treats or prevents these diseases. What is the definition of a non-communicable disease? Are certain diseases more common than others within your population? Does PA reduce the risk for some diseases more than others?

Methods & Results:

  • It seems that all the men dropped out of the study? Lines 94-95 state that there were 100 participants, 95 women, but then line 96 state, “all participants were Black African Women…” Why did all the men drop out? Does this affect your results? Why or why not?
  • I do not see “100 participants” in the Figure 1 flow chart? How and when did they drop out? This needs to be included in the flow chart.
  • A major omission from the methods is a detailed description of the Adapted Physical Activity Health Belief Model intervention. What specifically was done with regards to the intervention? How was each of the five components of the model addressed? What types of resources and information did you provide participants? How often did you consult participants? Was this in groups or individually? How long did these consultations last? How did you measure their perceptions and knowledge? What measures did you use and why?
  • Was the exercise intervention based on prior research or the American College of Sports Medicine (ACSM) handbook? How did you develop the exercise program?
  • Lines 111-112 stated that you measured compliance via rollcall and recording heart rates. How did you do this for the two exercise sessions performed at the participant’s home? What do you mean “by heart rate”? What device did you use to measure heart rate? Was there a targeted range that qualified as “compliance”? How many participants did or did not meet this criterion, and how did you handle that data?
  • You need to include a description and specifics (model, brand/manufacture) of all the equipment used in the study and you need to describe and cite all the self-report measures used in the study. Line 111 said you recorded heart rate, so how did you do this, and with what specific piece of equipment? How was physical activity measured, and what measure did you use? How was blood pressure measured, and what equipment did you use? How was blood glucose measured, and what device did you use? How was cholesterol measured, and what device did you use? Was this information self-reported? If so, what self-report measure did you use? How was PA knowledge measured, and what measure did you use? How was heart disease knowledge measured and what measure did you use? What were the reliability coefficients of the measure?
  • What was explicitly asked during the qualitative interviews? What questions were asked? Were the questions based on theory and prior research? What were the standardized responses and follow-up questions? How long did the interviews last? Was this performed individually or in groups? Who performed the interviews and when? Did you have multiple interviewers and transcript raters to determine the validity and reliability of the selected themes and subtexts? Did you perform triangulation (Renz et al., 2018)?
  • In Figure 1, how were the participants allocated to the focus group vs. only quantitative measures? How come you didn’t assess qualitative and quantitative variables from all the participants?
  • I understand that “mixed-methods” means applying quantitative and qualitative methods to ALL participant data. It is therefore, invalid for you to compare two different sample data sets simultaneously (i.e., one group of participants in your study had only quantitative data collected while another group had quantitative and qualitative data collected). You cannot treat these two groups as one sample in an analysis of qualitative and quantitative data if you don’t have qualitative data for most of the other participants. Therefore, as a “mixed-method” study, it is only appropriate to simultaneously analyze the participants who had BOTH qualitative and quantitative complete data (see - Timans et al., 2019).
  • For treatment intervention studies, if a considerable percentage of participants drop out (which your study had ~40% dropout, which is considerable), it is best practice to use the Intention To Treat analysis for the quantitative variables and thereby include all participant data in the final quantitative analyses (e.g., Last Observation Carried Forward (LOCF) method) (see - Gupta, 2011; McCoy, 2017). Since ~40% of your sample dropped out, an Intention To Treat analysis should be performed to determine if the high drop-out rate affected or biased your results. For example, might there be fundamental differences between the participants who remained in the intervention vs. those who dropped out? If only highly motivated participants remained in the study, for example, wouldn’t this reduce the study’s external validity?
  • Why were the qualitative themes “predetermined” (lines 183-194)? Who decided on the themes and the criteria for predetermining the themes? Usually, themes emerge from the post-hoc review of the participant transcripts (see – Singh & Richards, 2003). Do you have a citation to back your approach?
  • Figure 2 does not follow APA format (see - American Psychological Association, 2020)
  • In Table 3, I only see one p-value per NDC risk factor, yet there are THREE timepoints (therefore, I should see three p values for each NDC risk factor). Please include all three p values for each NCD risk factor corresponding to all three timepoint comparisons (e.g., baseline to 12 weeks, baseline to 24 weeks, 12 weeks to 24 weeks). Note that this should include Intention To Treat.
  • Why was a non-parametric test performed on PA knowledge scores? (line 296). Did the data violate a statistical assumption? If so, which assumption and what were the criteria you used to determine that the violation warranted a non-parametric test vs. a correction (e.g., Greenhouse-Geisser or Huynh-Feldt that you used for other tests)?

References

American Psychological Association. (2020). Publication manual of the American Psychological Association (7th ed.). https://doi.org/10.1037/0000165-000

Gupta, S. K. (2011). Intention-to-treat concept: a review. Perspectives in Clinical Research, 2(3), 109.

McCoy, C. E. (2017). Understanding the intention-to-treat principle in randomized controlled trials. Western Journal of Emergency Medicine, 18(6), 1075.

Renz, S. M., Carrington, J. M., & Badger, T. A. (2018). Two strategies for qualitative content analysis: An intramethod approach to triangulation. Qualitative health research, 28(5), 824-831

Singh, S., & Richards, L. (2003). Finding “central” themes in qualitative research. Qualitative Research Journal, 33(11), 5-17.

Timans, R., Wouters, P., & Heilbron, J. (2019). Mixed methods research: what it is and what it could be. Theory and Society, 48(2), 193-216.

Reviewer 2 Report

This manuscript, entitled “The influence of supervised exercise on perceptions and knowledge of non-communicable disease risk factors and physical activity among women from a low resourced setting: A mixed-methods study”, investigated the changes in the perceptions and knowledge of NCDs and PA during the course of the supervised exercise (baseline, 12 weeks and 24 weeks) using both qualitative and quantitative methods. The results showed that female participants enrolled in this study improved their knowledge of NCDs and PA after the supervised exercise. This study was theoretically sound and added novel findings to the existing evidence. The manuscript was well written. However, before this manuscript could be accepted for publication, there are some comments which need to be addressed.

My major concern is about the underlying mechanism why the supervised exercise program could change the participants’ perception or knowledge. It is not clear to me that if the supervisors of the exercise program have provided any information or education for the participants. If yes, was there a standardized process when the supervisors delivered the knowledge translation? On the other hand, if the supervisors simply provided the exercise intervention, how this approach may change the participants’ perception and knowledge? Although this study was theoretically rigorous as it was developed based on the HBM, the introduction should provide a rationale to address the potential benefits of the supervised exercise. In addition, more discussion is needed to describe the underlying mechanism how participation in the supervised exercise may change the perceptions and knowledge of NCDs and PA.

There are some minor comments.

  1. Line 29: [1,4]
  2. Line 46-55: The key information of this paragraph seems to be “female sex is a risk factor of physical inactivity and NCDs”. If I am correct, the first sentence may be irrelevant to this idea. I am wondering if it could be removed from this paragraph.
  3. Line 50: full name of SA
  4. Line 51: [15,16]
  5. Line 56-62: This paragraph is confusing. I am not sure if I could understand the key information addressed here.
  6. Line 60: [19.20]
  7. Line 63: Is the HBM is the same as the adapted PAHBM?
  8. Figure 1: I was surprised by the high dropout rate which was more than 50% at the end of the program and could significantly bias the results. I was wondering if the authors analyzed the reasons why so many participants withdrew from this study and compared the differences in quantitative data between those participants who stayed until the end and quitted the exercise program.
  9. Figure 1: n = 55, n = 44
  10. Line 107-108: In order to improve the reproductivity or application of this study, I would recommend to give some examples (i.e., types) of aerobic activities and resistance training.   
  11. Line 117: delete “.”
  12. Line 122: “24-week post intervention” may indicate the 24th week after the end of the intervention which would confuse the audience. “Of the intervention” may be better.
  13. Line 127: The full names of the abbreviations should be provided. Also, when there are various procedures of collecting data on PA, more details are needed to better describe how PA was measured and analyzed.
  14. Line 130, Line 138, Line 156: numeric subheadings
  15. Line 159: I would recommend to combine this section and qualitative data analysis.
  16. Table 3: As described in Figure 1, there should be 44 participants who have completed the quantitative assessments. However, the sample size for each variable did not match the information in Figure 1. More description is needed.
  17. Line 259: To be consistent, I would recommend to use either “partial eta squared” or “??2
  18. Section 3.3.3: As there are many questions in the survey, I am curious why the authors have selected the results of those specific questions to report. Is there any reason for this? Also, as the authors mentioned that “age is non-modifiable risk factor……”, I could not find this question in Table 4.
  19. Line 303: Table 4
  20. Line 307: 71%?
  21. Table 4 and 5: The legends should be changed. I realized that the number in the tables indicated the number and the percentage of participants who answered correctly. The “score” in the legends were misleading. Furthermore, I suggest the authors to go through all numbers to make sure that the information is correct. For example, in the second question of Table 5, the percentage should be 61 (58/95).
  22. Line 355: , while
  23. Line 356: are

Reviewer 3 Report

  • Thank you for the author's contribution to this study. The manuscript shows an interesting topic, but the introduction and methods parts need to revise. The result is well presented, and the conclusions are well structured.

    My detailed comments are presented here:

    • The Introduction is general and brief. Please introduce deeply the Health Belief Model (HBM). Furthermore, there are variables included in this study that should be highlight their importance in the introduction (e.g., the role of the cholesterol/blood pressure/BMI/etc. and why it's important in this study.)
    • Add more information on the physical activity readiness questionnaire (PAR-Q).
    • Description of the qualitative data are superficies, please explain more on that method.
    • How quantitative data were measured, especially blood pressure, blood glucose, cholesterol, and PA? I would recommend explaining this method in different sections or paragraphs.
    • It would be ideal to add a "measure" subchapter into the methods, since there sociodemographic and PAR-Q questionnaires were used. It would help to understand the results.

Reviewer 4 Report

This is a well-conducted study that should be highly relevant for readers of IJERPH. Reports from real-life interventions (rather than resource-intensive “proof of concept”-studies ) are in short supply. The study is especially relevant as the target group is in low resourced settings – lifestyle-wise the most vulnerable part of societies both in industrialized and emergent nations.

The manuscript is properly structured and well-written.

In a 24-week exercise-cum-education intervention 95 participants from low-resourced settings in South Africa improved both health-literacy, blood pressure and anthropometric measures of obesity. The title describes the article but I would suggest (Suggestion 1) to shorten it and give away some of the results to attract interested readers e.g. “24-week exercise intervention among women from low resourced setting: Improvement in health literacy and cardiometabolic health”. The abstract reflects the content of the article. The introduction provides a context and states the authors aims – (Suggestion 2) as a rule rephrasing these as a hypothesis gives better focus to an article. Methods are explained properly, and measurements described adequately. The authors are commended for the flow-chart that gives a good overview of the study – lower rates of dropout in the focus group (35% vs 64%) should be commented upon in the discussion. The result-section is clearly laid out and follows a logical sequence. Figures and tables are informative and appear accurate. However, I would suggest (Suggestion 3) replacing figure 2 (which partly duplicates data from table 3) with a similar figure comparing cardiometabolic profile of participants who remained in the study with dropouts at 12 and 24 weeks. Identifying possible baseline characteristics that predict dropout would enable us to design modified interventions for that particular group (in our experience those with the least favourable baseline data). Table 4 is difficult to understand – (Suggestion 4) either provide a caption explaining knowledge-score or give only proportions of correct answers. Table 6 is a very good example of how findings from mixed methods studies can be presented in a reader-friendly way! The discussion puts the results into proper perspective. I would suggest (Suggestion 5) to elaborate on the theme of high dropout-rates and what should/may be done to reach those particular groups – being part of a focus group apparently is favourable. The conclusion is supported by the results – again, (Suggestion 6) I would mention conclusions regarding adherence/dropout.

Summarizing a most interesting study providing valuable data on what can be expected from a lifestyle-intervention in the current setting both in terms of adherence and results. Very valuable data for both clinical practice and public health planning. The 6 suggestions mentioned above might contribute to further improvement of an already well-written manuscript.